# HEX: Merging Heavy-Hitters and Expanders for Adaptive KV Cache Optimization in Long-Context Inference

## Abstract

Key–Value (KV) caching accelerates large-language model inference but grows linearly with sequence length, quickly exhausting GPU memory. Existing compression strategies such as quantization, pruning, or sparsification shrink this footprint, but often degrade performance. Most pruning methods discard crucial connections and disrupt information flow, while dynamic heuristics often lack theoretical basis. We propose HEX, a cache compression strategy that is both structurally efficient and adaptive. HEX constructs a sparse backbone using expander graphs with spectral guarantees on connectivity, and augments it with heavy-hitter and recent tokens to capture input-specific context. The selected entries are stored in full precision, while the remaining cache is quantized to retain information at low cost. The expander masks are precomputed and static, thus significantly reducing computational overhead and aiding sparse implementations. Experiments on GSM8k, CoQA, TruthfulQA, and LongBench across models of varying sizes show that HEX consistently outperforms existing methods at higher compression rates without retraining. These results illustrate how principled eviction layouts grounded in graph structure and input dynamics can yield stronger accuracy–efficiency trade-offs for long-context inference even for limited cache budgets.

## 1 Introduction

Efficient inference in Large Language Models (LLM) is an important research challenge as deployment moves into resource-constrained and real-time environments. An essential optimization during inference is *Key-Value (KV) caching*, which stores the intermediate attention states of previously processed tokens and reduces the computational cost per token from $O(n^2)$ to $O(n)$, where $n$ is the total sequence length. Although caching enables fast generation, the cache itself grows linearly with sequence length, creating severe memory bottlenecks that limit throughput and prevent practical deployment in long-context tasks such as chain-of-thought reasoning, code generation, and document-level question answering.

In order to address this, prior work compresses KV caches using techniques like *quantization* (Liu et al., 2024d; Sheng et al., 2023), *low rank* representations (Chang et al., 2025; Lin et al., 2025), *layer-wise* and *head-wise compression* (Liu et al., 2024a; Ge et al., 2024c), and *pruning/eviction* (Xiao et al., 2024). Hybrid frameworks such as GEAR (Kang et al., 2024) combine these strategies, storing most entries in ultra-low precision, recovering residuals with a low-rank approximation, and preserving outliers through sparse masks.

Channel significance and the magnitude of attention weights are generally considered when pruning the cache (Xu et al., 2025; Lv et al., 2025); for instance, GEAR's sparse component relies on magnitude pruning, which introduces unstructured sparsity. Although effective in some inference tasks (Joo et al., 2025), unstructured pruning can remove crucial connections and degrade performance.

Structured sparsity methods address this by network aware pruning of the attention blocks at the depth (Liu et al., 2024b) or layer (Zhang et al., 2025b) level. These often lack principled theoretical guarantees. Viewed as a graph, the attention matrix represents connections among context tokens. Preserving connectivity in this graph is essential for proper information flow. In this work, we propose KV cache pruning using *structured sparsity via expander graphs*, which ensures that

each channel and token maintains multiple well-distributed connections throughout the network. Expander-based sparsity offers strong spectral guarantees of the masked attention matrix, improving connectivity, and preserving informative patterns even under aggressive compression. The pruning masks are precomputed using expander graph synthesis algorithms and do not have a run-time computational overhead.

While the expander mask provides a structured and theoretically grounded backbone, it is static and cannot adapt to the dynamic nature of input sequences. To address this, HEX augments the expander backbone with input-aware token selection. Among dynamic strategies, $H_2O$ (Zhang et al., 2023) is particularly suitable because of its simplicity and complementarity. $H_2O$ selects additional tokens using two intuitive principles: (i) heavy hitters, i.e., tokens that consistently receive high attention and are therefore more influential for downstream generation (Xu et al., 2025; Lv et al., 2025), and (ii) recency, i.e., the most recent tokens, which are empirically shown to play a dominant role in autoregressive decoding (Liu et al., 2024c; Xiao et al., 2024). By augmenting the static expander mask with these dynamically chosen tokens, HEX captures both long-range structural connectivity and short-term contextual relevance. These selected tokens are retained in full precision, while the remaining cache entries are quantized, allowing information to be preserved at minimal cost.

Experiments on GSM8k, TruthfulQA and LongBench benchmarks show that HEX outperforms existing baselines while achieving higher compression, without any retraining. Our experiments show that HEX achieves SOTA resutlts for reasoning tasks under very heavy compression (3 bit) and for long context tasks surpasses the full precision (Fp16) framework under 4 bit quantization.

## 2 RELATED WORK

**KV cache compression:** Reducing the memory footprint of the Key-Value (KV) cache has attracted significant attention as context length scales in LLM inference. *Quantization*-based methods reduce memory by storing cache tensors in low-bit formats. KIVI (Liu et al., 2024d) applies tuning-free 2-bit quantization with asymmetric treatment of keys/values. FLEXGEN (Sheng et al., 2023) formulates tensor placement as a linear programming problem, while KVTUNER (Li et al., 2025) searches for optimal precisions per layer. *Pruning and eviction* approaches discard less important tokens to maintain bounded cache sizes. STREAMINGLLM (Xiao et al., 2024) and H2O (Zhang et al., 2023) evict stale tokens, while TREEKV (He et al., 2025) and SNAPKV (Li et al., 2024) score importance via distance or attention statistics. SEPLLM (Chen et al., 2025) compresses between separators, and FASTGEN (Ge et al., 2024c) profiles heads for adaptive eviction.

KV states often admit compact bases. *Low-rank approximations* (PALU (Chang et al., 2025), MATRYOSHKAKV (Lin et al., 2025)) down-project hidden dimensions; LOKI (Singhania et al., 2024) scores tokens in a low-dimensional space. *Sparse representations* like dictionary-based methods (CSR (Zhang et al., 2025a), LEXICO (Kim et al., 2025)) achieve sparsity via learned or universal codebooks. Finally, *hybrid frameworks* combine multiple strategies. GEAR (Kang et al., 2024) integrates quantization, sparse outliers, and low-rank correction. LEANKV (Zhang et al., 2024), ROCKETKV (Behnam et al., 2025) mix eviction, sparse attention, and quantization. While hybrids achieve stronger trade-offs, they often lack proper guarantees and rely on heuristic allocations.

**Structured sparsity:** Most KV cache methods rely on magnitude pruning, which prioritizes extreme values but can overlook structurally important entries. Prior work in pruning and compression has shown that structured sparsity often yields better accuracy and hardware efficiency compared to unstructured pruning, due to its more balanced and regular coverage patterns (Wen et al., 2016; Evci et al., 2020).

Expander graphs are particularly attractive: they preserve connectivity under extreme sparsity, supported by well-established spectral guarantees (Marcus et al., 2015; Hoory et al., 2006). Recent work such as XoRA (Amaljith et al., 2025) has demonstrated their utility for efficient LLM finetuning, though their application to *KV cache compression* remains unexplored.

**Our Contribution:** We summarize our two core contributions below:

*(i)* **Precomputed expander-backed KV pruning:** We introduce a practical KV-cache pruning scheme based on precomputed $d$-regular expander-graph masks. The masks impose structural sparsity while provably preserving connectivity among token and channel coordinates, ensuring uniform

information flow under aggressive downsampling. The construction is theoretically grounded (spectral/sampling guarantees) and composes cleanly with existing compression primitives, providing a stable pruning backbone even for very long contexts.

*(ii)* **Augmentation of expander with a dynamic component:** To capture input-specific signal, we augment the static expander backbone with a light-weight dynamic policy that retains a small set of input-aware tokens (heavy-hitters and recent tokens). The resulting hybrid combines provable structural safety with adaptive fidelity, yielding a compression strategy that is both robust and precise in practice.

## 3 METHOD

We present our approach for compressing the key-value (KV) cache in transformer architectures by combining three complementary techniques: *quantization*, *static structural sparsity via expander-based selection*, and *dynamic heavy-hitter* selection with *recency* bias ($H_2O$). The central idea is to treat the KV cache as a heterogeneous memory where a small fraction of critical entries are preserved in full precision, while the remaining majority are aggressively compressed using low-bit quantization. This balances *memory efficiency*, *retention of important context*, and *connectivity* of tokens with a low overhead during decoding.

### 3.1 PROBLEM SETUP

We begin by formalizing the setting of interest: multi-head attention (MHA) and its grouped-query variant (GQA). For an input sequence of length $n$ with embeddings $X \in \mathbb{R}^{n \times d}$, queries, keys, and values are obtained via

$$Q = XW_Q, \quad K = XW_K, \quad V = XW_V,$$

with learned projections $W_Q, W_K, W_V \in \mathbb{R}^{d \times d}$. The hidden dimension $d$ is split across $h$ attention heads, each of size $d_h = d/h$ (MHA) or across fewer key–value heads (GQA).

For a query $q_t \in \mathbb{R}^{d_h}$ at position $t$, attention output is

$$\text{Attn}(q_t, K, V) \;=\; \text{softmax}\left(\frac{q_t K^\top}{\sqrt{d_h}}\right) V,$$

where $K, V \in \mathbb{R}^{n \times d_h}$ are the cached keys and values of the preceding $n$ tokens. During autoregressive decoding, these matrices accumulate over time to form the *KV cache*, which dominates memory usage for long contexts.

**Compression problem:** We seek a compression operator $\mathcal{C}$ that acts jointly on the key and value matrices:

$$\mathcal{C} : (K, V) \;\mapsto\; (\widehat{K}, \widehat{V}),$$

where $K, V \in \mathbb{R}^{n \times d}$ are the original cache matrices and $\widehat{K}, \widehat{V} \in \mathbb{R}^{n \times d}$ are the compressed outputs. The operator $\mathcal{C}$ reduces storage while ensuring that, for each query $q_t$, the deviation remains small.

$$\epsilon_t \;=\; \left\| \text{Attn}(q_t, K, V) - \text{Attn}(q_t, \widehat{K}, \widehat{V}) \right\|_2$$

**Assumptions:** *(i)* Autoregressive decoding is assumed, but the formulation is not tied to causality.

*(ii)* Compression is applied directly to $K$ and $V$ in their native $(n, d)$ form. The operator does not modify the attention weight pattern.

**Problem statement:** *Given key and value matrices $(K, V) \in \mathbb{R}^{n \times d}$ from attention, design a compression operator $\mathcal{C}$ that reduces storage while preserving attention fidelity, i.e., ensuring $\epsilon_t$ is small for all queries.*

### 3.2 STATIC STRUCTURAL SPARSITY VIA EXPANDERS

KV cache compression requires removing most entries while still preserving connectivity across tokens and channels. Unstructured pruning often produces highly clustered supports, so we use *expander graph*-based masks—sparse yet highly connected structures with strong spectral guarantees.

**Background:** We view the cache matrix $K \in \mathbb{R}^{n \times d}$ as the biadjacency of a bipartite graph $G = (U, V, E)$, where $U$ indexes the $d$ channel coordinates and $V$ indexes the $n$ token positions. A $(d_1, d_2)$-biregular expander is one where each token node in $V$ connects to exactly $d_1$ channels, and each channel node in $U$ connects to exactly $d_2$ tokens. This guarantees uniform coverage across both axes of the cache.

The quality of an expander is governed by its *spectral gap*: the difference between the largest and second-largest eigenvalues of its adjacency matrix. Intuitively, a larger gap means that no subset of tokens or channels can become isolated, ensuring that information flows evenly across the sparse structure. Near-optimal expansion is achieved by *Ramanujan graphs*, which satisfy the celebrated bound (Hoory et al., 2006; Marcus et al., 2015):

$$\lambda_2 \leq \sqrt{d_1 - 1} + \sqrt{d_2 - 1},$$

where $\lambda_2$ is the second-largest eigenvalue. This condition certifies that expander-based masks achieve the best possible trade-off between sparsity and connectivity. Additional details on spectral expansion and constructions of Ramanujan graphs are provided in Appendix A.2 and Table 5.

**Key properties for KV caching:** Expander-based masks provide three structural guarantees that are directly relevant to compression. We give the intuition here and defer full spectral statements and proofs to Appendix A.1.

*(i)* **Uniform retention:** Expanders ensure that no subset of tokens or channels is disproportionately discarded. In practice, this means every region of the cache maintains a balanced share of active entries, avoiding "holes" where entire segments of context would otherwise vanish.

*(ii)* **Spectral contraction:** The spectral gap of expanders guarantees that any perturbation introduced by masking or quantization is damped rather than amplified when propagated through attention. For the cache, this provides a uniform stability guarantee: errors remain bounded even under aggressive compression.

*(iii)* **Rapid mixing:** Expanders distribute information evenly: any local error or loss of detail is quickly spread across the cache rather than remaining confined to a single region. For KV compression, this means quantization noise or sparsity artifacts are smoothed out, making the resulting cache more robust for downstream attention.

**Implementation Details:** Expander masks $Mask_{\exp} \in \{0, 1\}^{n \times d}$ are static: they depend only on the sequence length $n$, hidden dimension $d$, and sparsity $\rho$. To make them practical, we pre-compute Ramanujan expanders (for their optimal spectral guarantees) and store the resulting masks in compressed sparse row (CSR) format[1], thus adding no runtime overhead in mask computation unlike most other pruning techniques. Because datasets often contain queries of similar length, we keep a small LRU cache of recently used masks in memory, which avoids repeated disk access. If a new $(n, d, \rho)$ configuration is encountered, an expander is generated on the fly using a lightweight constructive routine (details in the Appendix A.2). The generation time is minimal and is analyzed in the Appendix A.3. In practice, this yields efficiently retrievable masks that integrate seamlessly into the compression pipeline.

## 3.3 DYNAMIC SELECTION OF HEAVY HITTERS ($H_2O$)

While static expander masks guarantee structural coverage, they remain agnostic to the actual distribution of attention at inference time. To adaptively preserve the most predictive parts of the context, we introduce a framework similar to *Heavy Hitter Oracle($H_2O$)* (Zhang et al., 2023). $H_2O$ complements the static structure by dynamically identifying tokens that are either highly influential or recently generated.

**Heavy hitters:** A natural measure of token importance is the attention mass it attracts, a criterion widely used in prior work on importance-based compression and pruning (e.g. Michel et al., 2019).

---

[1]CSR is preferred over COO/CSC since it allows both compact storage and fast row access when converting to device tensors.

Given attention weights $A_{i,t}^{(q)}$ from head $i$ at query position $q$, we define the cumulative score for token $t$ as

$$s_t = \sum_{i=1}^{h} \sum_{q=1}^{Q} A_{i,t}^{(q)}.$$

Unlike the original $H_2O$, which selects heavy hitters separately per head, our approach identifies the most important tokens globally, ensuring that the KV cache preserves the dominant context for downstream predictions.

**Recency:** Autoregressive decoding exhibits a strong locality bias: the most recent tokens often dominate the prediction of the next token. Retaining a small fixed window of the latest $r$ tokens is therefore a widely adopted strategy in long-context compression (Ge et al., 2024a). We denote this window-based retention as $Mask_{\text{local}}$, which guarantees coverage of immediate local context.

**Combined mask:** The final dynamic mask is obtained as the union

$$Mask_{\text{H}_2\text{O}} = Mask_{\text{hh}} \vee Mask_{\text{local}}.$$

### 3.4 COARSE GRAINED QUANTIZATION

After selecting entries with the static expander and dynamic $H_2O$ masks, the remaining KV cache entries can be compressed to reduce memory usage. We adopt an asymmetric, coarse-grained Key-Channel / Value-Token (KCVT) quantization strategy (Liu et al., 2024d; Ge et al., 2024b).

**Asymmetric quantization:** Attention matrices contain both small and extreme values, and symmetric quantization can bias zeros or truncate extremes, degrading attention quality. Asymmetric min–max quantization maps values to a discrete range based on the actual minimum and maximum of each group, ensuring that zeros and large values are preserved:

$$\tilde{X}_{i,j} = \text{round}\Big((X_{i,j} - \min_g X)/\Delta_g\Big) \cdot \Delta_g + \min_g X,$$

where $\Delta_g$ is the dynamic range of group $g$.

**Role-aware grouping: keys vs. values:** Keys determine attention affinities across channels, while values carry the information to be aggregated over tokens. To preserve their respective roles, keys are quantized along the channel dimension and values along the token dimension (Liu et al., 2024d). This coarse-grained grouping captures the most critical variations while allowing aggressive compression, and is simpler and more hardware-efficient than fine-grained alternatives that split groups into multiple smaller blocks.

### 3.5 INFERENCE PIPELINE

Figure 1 illustrates the complete KV cache compression and inference workflow. Given the query, key, and value matrices $(Q, K, V)$ for a new input token, HEX proceeds in three steps:

**1. Static Structural Masking:** A precomputed expander mask $Mask_{\text{exp}}$ identifies a subset of cache entries that preserve connectivity across channels and tokens. These entries are retained in full precision to ensure robust information flow.

**2. Dynamic Input-Aware Selection:** The $H_2O$ mechanism dynamically selects tokens that are either heavy hitters—accumulating high attention mass across heads and query positions—or among the most recent tokens. This produces the mask $Mask_{\text{H}_2\text{O}}$, whose entries are also retained in full precision to capture input-specific context.

**3. Quantization:** The masks $Mask_{\text{exp}}$ and $Mask_{\text{H}_2\text{O}}$ are combined using a logical OR. Entries indicated by the combined mask are stored in full precision, while all remaining entries are quantized using coarse-grained KCVT with asymmetric min–max scaling. Keys are quantized along channels, and values along tokens, reflecting their distinct roles in attention computation.

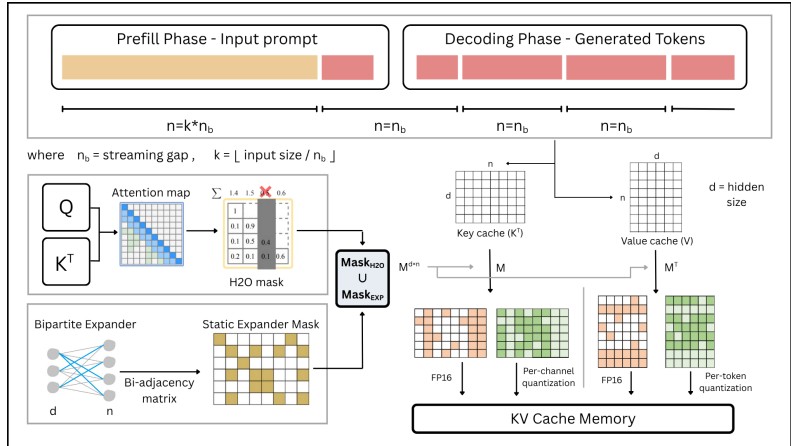

Figure 1: The complete KV cache compression and inference workflow (with append mode) for HEX. The notation $\lfloor \cdot \rfloor$ represents the floor operation.

## 4 EXPERIMENTS

### 4.1 EXPERIMENTAL SETUP

We evaluate our method, HEX, on four open-weight models: LLaMA-3 8B, Mistral-7B, LLaMA-2 13B, and LLaMA-2 7B. Three reasoning and QA benchmarks are considered—GSM8k (mathematical reasoning), TruthfulQA (factual consistency), and CoQA (conversational QA)—alongside LongBench (long-context evaluation). GSM8k is evaluated with the standard 8-shot prompt, reporting exact match accuracy. TruthfulQA is evaluated in the generation setting of lm-eval-harness with its default 6-shot prompt, reporting BLEU-based accuracy. CoQA is also evaluated using lm-eval-harness with exact match accuracy on the test split. LongBench is evaluated in its default zero-shot setting with per-subset metrics as recommended in its benchmark suite (further details are provided in the Appendix). All experiments are run on an NVIDIA H100 GPU (80GB). We simulate KV cache compression without modifying the underlying model parameters.

**Streaming Inference Setup:** We adopt a streaming inference setup for applying KV cache compression during decoding. Let the streaming block size be denoted by $n_b = 96$. The prefix prompt is first partitioned into the largest multiple of $n_b$, which is immediately compressed, while any remainder is placed in a residual buffer. During generation, newly decoded tokens first fill this residual buffer until it reaches length $n_b$. Once full, the block is compressed and merged into the KV cache, after which decoding proceeds in blocks of size $n_b$. This design ensures that both the prompt and the generated tokens are consistently aligned to block boundaries, avoiding repeated mask resizing and yielding a more efficient pipeline. Tokens inside the residual buffer are maintained in full precision (fp16) until the block fills, at which point they are compressed. For compressing each block, we consider two update strategies:

*(i)* **Append mode:** each new compressed block is appended to the existing compressed KV cache without reprocessing earlier blocks.
*(ii)* **Full mode:** the entire KV cache is recompressed at the end of each block.

In our ablation experiments, we found that append mode achieves virtually identical accuracy to full mode for generation lengths up to 512 tokens, while requiring substantially less computation. Therefore, append mode is adopted as the default setting. For longer generations, full mode or hybrid approaches (e.g., periodic recompression after several blocks) can be used if desired.

### 4.2 RESULTS OF REASONING AND QA BENCHMARKS

We first evaluate HEX on three reasoning and QA tasks: GSM8k, TruthfulQA, and CoQA, using LLaMA-3 8B, LLaMA-2 13B, and Mistral-7B (v0.1). Baselines include the uncompressed FP16 model, KCVT quantization (3-bit coarse-grained KIVI), and GEAR (combining quantization, low-

Table 1: Performance on Reasoning and QA Benchmarks. *KV Size* is the average cache size relative to FP16. GEAR ($s = 2\%, r = 4$) is included as a baseline. TruQA here represents the TruthfulQA dataset. Best results are in **bold**.

| Model | | | LLaMA3-8B | | | LLaMA2-13B | | | Mistral-7B | | |
|---|---|---|---|---|---|---|---|---|---|---|---|
| Method | Bit $b$ | Avg KV size | GSM8k Acc | CoQA Acc | TruQA BLEU | GSM8k Acc | CoQA Acc | TruQA BLEU | GSM8k Acc | CoQA Acc | TruQA BLEU |
| FP16 | 16 | 100% | 53.98 | 67.35 | 43.45 | 29.64 | 66.37 | 30.23 | 42.61 | 67.30 | 41.00 |
| KCVT Quant | 3 | 20.75% | 32.22 | 64.28 | 32.68 | 17.97 | 62.27 | **32.80** | 34.87 | 63.27 | 35.01 |
| GEAR | 4 | 31.00% | 54.44 | **68.00** | 42.35 | 29.57 | **66.88** | 30.84 | 41.85 | **67.67** | 40.39 |
| HEX-3 | 3 | 25.35% | **55.42** | 66.97 | **42.47** | **29.64** | 65.22 | 29.50 | 41.62 | 65.00 | **42.72** |
| HEX-4 | 4 | 31.50% | 55.34 | 66.98 | 40.88 | 28.96 | 66.23 | 29.87 | **41.92** | 66.83 | 41.37 |

rank, and magnitude pruning). HEX operates in streaming append mode with a block size of 96 tokens: each block is stored in FP16 until filled, then compressed. Unless otherwise stated, HEX uses 3.125% expander sparsity, 2% heavy hitters, and a small recent window (8, 16 tokens). Datasets like GSM8k which require reasoning benefit from a longer recent window, while with all remaining entries quantized to 3 bits (HEX-3) or 4 bits (HEX-4). For GSM8k and CoQA, performance is measured by exact match accuracy; for TruthfulQA, we follow LM-Eval Harness and report BLEU-based accuracy. To contextualize accuracy relative to efficiency, we also report the effective KV cache size for all methods.

From Table 1 we observe that both HEX-3 and HEX-4 match or surpass the FP16 and GEAR baselines across multiple datasets. Notably, HEX-3—despite operating at an aggressive 3-bit compression—outperforms all baselines on 4 of 9 tasks and exceeds KCVT (3-bit) in most cases, demonstrating the effectiveness of HEX for reasoning and QA tasks under stringent sparsity constraints.

Table 2: Comparison of HEX with KIVI 2-bit and 4-bit compression on LLaMA-2 13B across GSM8k, CoQA, and TruthfulQA.

| Method | Bit $b$ | Ave. KV size | GSM8k Acc | CoQA Acc | TruQA BLEU |
|---|---|---|---|---|---|
| FP16 | 16 | 100% | 29.64 | 66.37 | 30.23 |
| KCVT Quant | 3 | 20.75% | 17.97 | 62.27 | 32.80 |
| KIVI-4 | 4 | 34.20% | 23.65 | 66.38 | 29.49 |
| KIVI-2 | 2 | 21.70% | 20.77 | 66.23 | 29.84 |
| HEX3$_{(KCVT)}$ | 3 | 25.35% | 29.64 | 65.22 | 29.50 |
| HEX4$_{(KCVT)}$ | 4 | 31.50% | 28.96 | 66.23 | 29.87 |

## 4.3 RESULTS ON LONG-CONTEXT BENCHMARKS

In our second set of experiments, we evaluate HEX on LongBench, a suite of long-context understanding tasks, using LLaMA-2 7B. All evaluations are conducted in the streaming inference setup (Section 3.1). We primarily compare against the FP16 model to establish the upper bound, and report the results for both HEX-3 and HEX-4 with expander sparsity 4.69% and heavy hitter ratio 2%. We follow the default LongBench evaluation protocol and provide task-specific details and dataset statistics in the Appendix A.4. This evaluation highlights how HEX performs in settings with substantially longer input and output sequences, demonstrating its robustness beyond standard reasoning and QA tasks.

Table 3 summarizes performance on the LongBench suite. As expected, HEX-3 (3-bit quantization) exhibits a larger performance gap relative to the FP16 baseline on long-context tasks than it does on the reasoning and QA benchmarks. This degradation aligns with the increased difficulty of preserving information over very long contexts under aggressive compression. Even so, HEX-3

Table 3: The results of Llama-3-8B-Instruct with HEXon LongBench. Best results are in **bold**.

| | NarrQA | Qasper | MFQA-en | MFQA-zh | HotpotQA | 2WikiMQA | MuSiQue |
|---|---|---|---|---|---|---|---|
| FP16 | 17.30 | 9.08 | 22.37 | 19.33 | 8.24 | 10.00 | 4.27 |
| w./ HEX3 | **18.90** | 7.50 | 18.21 | 17.52 | 8.02 | 8.03 | 3.66 |
| w./ HEX4$_{append}$ | 17.18 | 8.76 | **22.74** | 20.61 | 8.02 | 10.00 | **4.74** |
| w./ HEX4$_{full}$ | 17.54 | 8.95 | 22.15 | **20.78** | 7.76 | **10.47** | **4.74** |

| | DuReader | GovReport | QMSum | MultiNews | VCSUM | TRec | TriviaQA |
|---|---|---|---|---|---|---|---|
| FP16 | 23.16 | 26.82 | 20.66 | 5.82 | 9.91 | 63.00 | 84.92 |
| w./ HEX3 | 21.37 | 19.39 | 20.66 | 8.09 | 6.85 | 54.50 | 83.19 |
| w./ HEX4$_{append}$ | **23.59** | 26.90 | 21.11 | **6.80** | 8.46 | 60.50 | **85.17** |
| w./ HEX4$_{full}$ | 22.33 | **26.85** | **21.19** | 6.76 | 9.59 | 62.50 | 84.83 |

| | SAMSum | LSHT | PCount | PR-en | PR-zh | LCC | RBench | KV size | Avg. Acc |
|---|---|---|---|---|---|---|---|---|---|
| FP16 | 41.46 | 20.25 | 1.50 | 5.77 | 8.00 | 58.70 | 62.30 | 100% | 24.90 |
| w./ HEX3 | **42.29** | 18.50 | **2.75** | 4.92 | 7.50 | 54.78 | 58.86 | 26.45% | 23.12 |
| w./ HEX4$_{append}$ | 40.62 | 20.00 | 1.10 | **5.92** | **10.12** | 58.11 | 62.00 | 32.70% | 24.88 |
| w./ HEX4$_{full}$ | 42.26 | 20.00 | 2.50 | 5.58 | 7.75 | 57.92 | 61.98 | 32.70% | **24.97** |

delivers a competitive trade-off: given its high compression ratio, the absolute drop in average accuracy is modest, and on a number of individual datasets HEX-3 surprisingly matches or exceeds the FP16 baseline and outperforms HEX-4, an outcome we attribute to dataset-specific attention patterns where a compact, well-chosen set of preserved entries suffices for prediction.

HEX-4 (4-bit quantization) is substantially more robust: in our experiments it preserves average accuracy close to FP16 in the streaming append setting and exceeds FP16 when run in the full recompression mode. We interpret this behavior as follows. The append mode compresses each block once when it fills, which can break some cross-block connectivity; the full mode periodically recompresses the entire cache and therefore better preserves the expander backbone and its cross-block connections. Because expander-based masks are designed to maintain distributed connectivity, their benefit is most apparent when the cache is recompressed as a whole, which explains why HEX-4 gains relative to FP16 are largest in the full mode. Together, these results show that marginally higher bit-width (4 bits) plus a connectivity-preserving recompression schedule recovers nearly all task performance while still providing substantial memory savings.

### 4.4 ABLATION STUDIES

To better understand the contributions of different components of HEX, we conduct a set of ablation studies and hyperparameter sensitivity analyses on LLaMA-3 8B evaluated on the GSM8k benchmark. For the ablations, we test both 3-bit and 4-bit variants of HEX after selectively removing key components in order to quantify the impact of each mechanism on accuracy and compression. Beyond ablations, we study the sensitivity of HEX to different hyperparameter choices in Appendix A.5. Specifically, we vary the streaming gap size and compare append vs. full recompression modes, evaluate the effect of different quantization bit-widths, explore different H2O configurations (heavy hitter ratio and recent window size), and vary the expander sparsity levels.

**Ablations:** We first perform ablations on both the **3-bit** and **4-bit** variants of HEX. In each case, we selectively remove one or more components: (i) without the expander, (ii) without H2O, and (iii) without both expander and H2O. This setup allows us to quantify the relative contribution of each mechanism to overall accuracy and memory savings. Results are summarized in Table 4, showing that both components provide measurable improvements, and that removing both leads to the largest degradation.

The ablation results clearly show that all components of HEX are necessary and complementary. At both 3-bit and 4-bit, the full HEX consistently achieves the best accuracy, while removing either

Table 4: Ablation studies on GSM8k with LLaMA-3 8B. All experiments use streaming append mode with block size $n_b = 96$. "HEX (full)" denotes Expander + H2O active. "–" indicates that the corresponding component was not used. Rows marked with **EO** (*Equal Overhead*) indicate that the expander-only and H2O-only settings were tuned so that the overall fraction of FP16-preserved tokens approximately matches that of HEX, ensuring a fair comparison.

| Quant bits | Expander sparsity (%) | Heavy hitters (%) | Recent window | Accuracy (%) |
|---|---|---|---|---|
| **3-bit** | | | | |
| HEX (full) | 3.125 | 2 | 8 | **55.42** |
| Expander only | 3.125 | – | – | 34.95 |
| H2O only | – | 2 | 8 | 53.82 |
| Quant only | – | – | – | 32.60 |
| **4-bit** | | | | |
| HEX (full) | 4.6875 | 2 | 8 | **54.66** |
| Expander only | 4.6875 | – | – | 50.76 |
| Expander (EO) | 7.8125 | – | – | 50.64 |
| H2O only | – | 2 | 8 | 53.98 |
| H2O only (EO) | – | 5 | 16 | 53.53 |
| Quant only | – | – | – | 51.78 |

the expander or H2O causes notable drops. Expander-only struggles, especially at 3-bit, proving that static structure alone is insufficient, while H2O-only improves over expander-only but still falls short of HEX, showing that dynamic selection benefits from structural safety. Quantization-only performs worst, confirming the need for guided preservation. The EO (Equal Overhead) results further demonstrate that HEX's gains are not due to token budget but to the synergy of expander guarantees with dynamic token retention.

## 5 CONCLUSIONS, LIMITATIONS AND FUTURE DIRECTIONS

Our study demonstrates that expander graphs provide a principled and robust backbone for KV cache compression. Their spectral guarantees confer structural advantages over unstructured magnitude pruning, and our experiments show that this approach achieves competitive or improved accuracy while maintaining significant memory savings.

We intend to avoid the pathological quantization and dequantization overhead by fusing dequantization directly into the attention matmul kernel, so no full-precision tensors are ever materialized. It follows the same systems principle used in GEAR but is lighter, removing the low-rank stage, using coarse-grained KCVT quantization even at 3-bit, and amortizing quantization over larger 96-token blocks. These design choices eliminate the costly sequence of separate quantization, dequantization, and matmul operations.

Despite these benefits, our work has two main limitations. First, while we provide analytical estimates of KV-cache savings, empirical results on inference latency and throughput are not yet included; these will be incorporated in the revised version. Second, the achievable sparsity levels of expanders are tied to the underlying matrix dimensions, which restricts flexibility in selecting arbitrary sparsity targets. While the available levels are sufficiently granular for most practical applications, this constraint may limit fine-grained tuning.

Looking ahead, a promising direction is hardware acceleration. Expander masks exhibit highly regular connectivity patterns, unlike unstructured pruning, making them well suited for parallelization on modern accelerators. Custom kernels or sparsity-aware compilers could exploit this regularity to reduce memory access overheads and improve throughput. Hardware-friendly implementations of HEX could therefore unlock larger gains in both efficiency and scalability, particularly for long-context inference.

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

# A APPENDIX

## A.1 EXPANDER GRAPHS

An expander graph is a sparse graph that has strong connectivity properties, quantified using vertex, edge or spectral expansion. Intuitively, it is is a finite, undirected multigraph in which every subset of the vertices that is not "too large" has a "large" boundary. This can be quantified using the notion of Cheeger constants.

**Definition 1** (Expander and Cheeger constant). *A graph $\Gamma = (V, E)$ is an $\epsilon$-vertex expander if for every non-empty subset $X \subset V$ with $|X| \leq \frac{|V|}{2}$, we have $\frac{|\delta(X)|}{|X|} \geq \epsilon$, where $\delta(X)$ denotes the outer vertex boundary of $X$ i.e., the set of vertices in $\Gamma$ which are connected to a vertex in $X$ but do not lie in $X$. As $X$ runs over all subsets of $V$, the infimum of $\frac{|\delta(X)|}{|X|}$ satisfying the conditions above is known as the vertex Cheeger constant and is denoted by $\mathbf{h}_V(\Gamma)$.*

Given a graph with a large Cheeger constant, it is difficult to "separate," meaning that it is hard to isolate any subset of vertices from the rest of the graph without cutting many edges. This property facilitates the free flow of information across the entire network and is also known as expansion of a graph and the best expanders are the Ramanujan graphs.

Graphs are said to be *spectral expanders* if their spectral properties imply strong connectivity and expansion. Specifically, a graph $\Gamma$ is considered a spectral expander if the quantities $|\lambda_1 - \lambda_2|$ and $|\lambda_1 - \lambda_k|$ are bounded away from zero, where $k = n - 1$ for bipartite graphs and $k = n$ otherwise. These bounds on the spectral gap ensure that the graph exhibits good expansion properties, as the eigenvalues provide insight into the graph's ability to resist being "cut" into disconnected components.

For a graph to be $d$-regular implies that all vertices have the same degree, which contributes to its regular structure. In the special case of *bipartite graphs*, the graph is said to be $d$-regular if it consists of two sets of vertices, where every vertex in each set has exactly $D$ edges connected to vertices in the other set, maintaining equal degrees across both partitions.

## A.2 GENERATION OF EXPANDER MASKS

Given an $(n_1, n_2)$ complete bipartite graph, we generate a good expander mask for it. According to the discussion in the previous section, we wish to ensure that this mask has a low degree (in this case $(d_1, d_2)$ bi-degree with $n_1 d_1 = n_2 d_2$ and high Cheeger constant). This brings us to the notion of Ramanujan masks. A Ramaunjan graph is an extremal expander graph in the sense that its spectral gap (and hence also the Cheeger constant) is almost as large as possible. Here, we shall be concerned with bipartite Ramanujan graphs. Recall that a bi-partite graph is said to be balanced if the number of vertices in each of the partitions are the same and it is said to be unbalanced otherwise.

**Definition 2** (Bipartite Ramanujan graphs). *Let $\Gamma = (V, E)$ be a $d$-regular ($d \geq 3$) balanced bipartite graph. Let the eigenvalues of its adjacency matrix be $\lambda_n \leq \lambda_{n-1} \leq \ldots \leq \lambda_2 \leq \lambda_1$. Then $\Gamma$ is said to be Ramanujan iff $|\lambda_i| \leq 2\sqrt{d-1}$, for $i = 2, \ldots, (n-1)$. For an unbalanced $(d_1, d_2)-$biregular bipartite graph ($d_1, d_2 \geq 3$), the condition of being Ramanujan changes to $|\lambda_i| \leq \sqrt{d_1 - 1} + \sqrt{d_2 - 1}$, for $i = 2, \ldots, (n-1)$.*

A detailed description of Ramanujan graphs can be found in (Hoory et al., 2006, sec. 5.3). One can generate the expander (Ramanujan) masks through the following two approaches.

1. Deterministic generation using LPS construction and Ramanujan $r$-coverings.
2. Random generation of bi-regular bipartite graphs and checking for Ramanujan criteria.

To generate expander masks deterministically, the Lubotzky–Phillips–Sarnak (LPS) construction is a good choice, which produces Ramanujan graphs known for their optimal expansion properties. The LPS method constructs $(p + 1)$-regular graphs using quaternion algebras over number fields, where $p$ is a prime satisfying certain congruence conditions. These graphs exhibit excellent spectral characteristics, making them ideal candidates for our masks. Additionally, Ramanujan $r$-coverings involve creating larger Ramanujan graphs from smaller ones through covering projections while preserving their expansion properties. This approach ensures that the resulting bipartite graphs have the desired bi-degree $(d_1, d_2)$ and high Cheeger constant, which is crucial for the effectiveness of the masked weight matrices in transformers.

We adopt a computationally flexible, *randomized generation approach* that produces $(d_1, d_2)$-bi-regular bipartite graphs of arbitrary sizes and filters them using spectral criteria. Explicit constructions such as LPS graphs or Ramanujan $r$-coverings impose arithmetic and regularity constraints that are incompatible with the general $(n_1, n_2, d_1, d_2)$ settings required for transformer architectures. Randomized generation avoids these restrictions while still allowing us to enforce the Ramanujan bound exactly.

Our method proceeds as follows. We first sample a $(d_1, d_2)$-biregular bipartite graph using a configuration-model - style pairing: each vertex in the left and right partitions is assigned $d_1$ and $d_2$ half-edges respectively, and a uniform random perfect matching between half-edges is drawn. This guarantees biregularity and the condition $n_1 d_1 = n_2 d_2$ by construction. The resulting adjacency matrix is then evaluated spectrally, and the candidate is accepted only if all nontrivial eigenvalues satisfy the Ramanujan bound. If the bound is violated, the sampling step is repeated. In practice, acceptance rates are high for the degree ranges relevant to our models, making the procedure computationally efficient.

The accepted adjacency matrix is subsequently used as the expander mask for the transformer layer. This randomized-and-filtered construction yields masks with strong spectral expansion while accommodating arbitrary layer dimensions and degree choices.

## A.3 TIME FOR EXPANDER MASK GENERATION

We empirically measure the time required to generate connected $(d_1, d_2)$-biregular bipartite expanders of size $d \times n$ (where $d$ is hidden dimension and $n$ is sequence length) using our vectorized

Table 5: Analysis of Ramanujan bound and spectral gap of expanders used. The sparsity for all expanders here is: $s = 3.125\%$. $\lambda_1$ is the largest and $\lambda_2$ is the second-largest eigenvalue.

| Size | $d_m$ | $d_n$ | $\lambda_1$ | $\lambda_2$ | Ramanujan Bound | Spectral Gap |
|------|-------|-------|-------------|-------------|-----------------|--------------|
| 1024_96 | 3 | 32 | 9.8 | 6.83 | 6.98 | 2.96 |
| 1024_192 | 6 | 32 | 13.86 | 7.69 | 7.8 | 6.16 |
| 1024_288 | 9 | 32 | 16.97 | 8.19 | 8.4 | 8.78 |
| 1024_384 | 12 | 32 | 19.6 | 8.72 | 8.88 | 10.87 |
| 1024_480 | 15 | 32 | 21.91 | 9.11 | 9.31 | 12.8 |
| 1024_576 | 18 | 32 | 24 | 9.44 | 9.69 | 14.56 |
| 1024_672 | 21 | 32 | 25.92 | 9.84 | 10.04 | 16.08 |
| 1024_768 | 24 | 32 | 27.71 | 10.16 | 10.36 | 17.55 |
| 1024_864 | 27 | 32 | 29.39 | 10.5 | 10.67 | 18.9 |
| 1024_960 | 30 | 32 | 30.98 | 10.65 | 10.95 | 20.34 |
| 1024_1152 | 36 | 32 | 33.94 | 11.24 | 11.48 | 22.7 |
| 1024_1248 | 39 | 32 | 35.33 | 11.49 | 11.73 | 23.83 |
| 1024_1536 | 48 | 32 | 39.19 | 12.2 | 12.42 | 26.99 |
| 1024_1728 | 54 | 32 | 41.57 | 12.61 | 12.85 | 28.96 |
| 1024_1920 | 60 | 32 | 43.82 | 13.03 | 13.25 | 30.79 |
| 1024_4032 | 126 | 32 | 63.5 | 16.48 | 16.75 | 47.02 |
| 1024_6528 | 204 | 32 | 80.8 | 19.45 | 19.82 | 61.35 |
| 1024_8160 | 255 | 32 | 90.33 | 21.1 | 21.51 | 69.23 |

construction procedure. As shown in Fig. 2, the generation time grows smoothly with expander size and remains well within practical limits: under $0.15$ seconds for moderate expanders ($d = 1024$) and below 3 seconds even for larger sizes ($d = 4096$). This efficiency arises from a fully vectorized stub-pairing step, fast duplicate elimination and localized double-edge swaps that enforce simplicity and connectivity without requiring any costly global reconstruction. These optimizations make on-the-fly generation of high-quality expanders sufficiently fast to impose negligible overhead during training or inference.

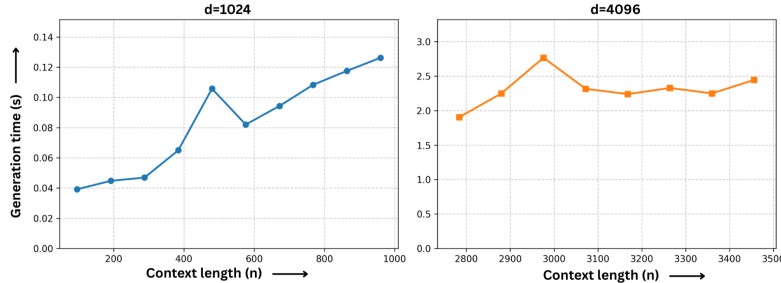

Figure 2: Expander generation time analysis for varying dimensions.

## A.4 LONGBENCH DATASET STATISTICS

Table 6 presents the statistics of the LongBench dataset, including task categories, sources, average input lengths, evaluation metrics, languages, and the number of examples used in our experiments. For our experimental setup, the input context length was truncated from the middle to fit within the models' maximum context lengths.

## A.5 HYPERPARAMETER SENSITIVITY.

Next, we study the sensitivity of HEX to its hyperparameters:
*(i)* Streaming gap size and method: we vary the block size used for streaming inference and compare the default *append* mode against the *full* recompression mode.
*(ii)* $H_2O$ parameters: we vary both the heavy-hitter ratio and the size of the recent-token window, to

Table 6: Overview of LongBench dataset statistics.

| Dataset | ID | Source | Avg len | Metric | Language | #data |
|---|---|---|---|---|---|---|
| **Single-Document QA** | | | | | | |
| NarrativeQA | 1–1 | Literature, Film | 18,409 | F1 | English | 200 |
| Qasper | 1–2 | Science | 3,619 | F1 | English | 200 |
| MultiFieldQA-en | 1–3 | Multi-field | 4,559 | F1 | English | 150 |
| MultiFieldQA-zh | 1–4 | Multi-field | 6,701 | F1 | Chinese | 200 |
| **Multi-Document QA** | | | | | | |
| HotpotQA | 2–1 | Wikipedia | 9,151 | F1 | English | 200 |
| 2WikiMultihopQA | 2–2 | Wikipedia | 4,887 | F1 | English | 200 |
| MuSiQue | 2–3 | Wikipedia | 11,214 | F1 | English | 200 |
| DuReader | 2–4 | Baidu Search | 15,768 | Rouge-L | Chinese | 200 |
| **Summarization** | | | | | | |
| GovReport | 3–1 | Government report | 8,734 | Rouge-L | English | 200 |
| QMSum | 3–2 | Meeting | 10,614 | Rouge-L | English | 200 |
| MultiNews | 3–3 | News | 2,113 | Rouge-L | English | 200 |
| VCSUM | 3–4 | Meeting | 15,380 | Rouge-L | Chinese | 200 |
| **Few-shot Learning** | | | | | | |
| TREC | 4–1 | Web question | 5,177 | Accuracy (CLS) | English | 200 |
| TriviaQA | 4–2 | Wikipedia, Web | 8,209 | F1 | English | 200 |
| SAMSum | 4–3 | Dialogue | 6,258 | Rouge-L | English | 200 |
| LSHT | 4–4 | News | 22,337 | Accuracy (CLS) | Chinese | 200 |
| **Synthetic Task** | | | | | | |
| PassageCount | 5–1 | Wikipedia | 11,141 | Accuracy (EM) | English | 200 |
| PassageRetrieval-en | 5–2 | Wikipedia | 9,289 | Accuracy (EM) | English | 200 |
| PassageRetrieval-zh | 5–3 | C4 Dataset | 6,745 | Accuracy (EM) | Chinese | 200 |
| **Code Completion** | | | | | | |
| LCC | 6–1 | Github | 1,235 | Edit Sim | Python/C#/Java | 500 |
| RepoBench-P | 6–2 | Github repository | 4,206 | Edit Sim | Python/Java | 50 |

measure their effect on preserving attention to critical tokens.

*(iii)* Expander sparsity: we test different sparsity levels to quantify how aggressively the expander can prune while maintaining accuracy.

Results are presented in Figure 3. Overall, we find that HEX is stable across a wide range of configurations. Append streaming mode is nearly indistinguishable from full recompression for moderate sequence lengths, making it a practical default.

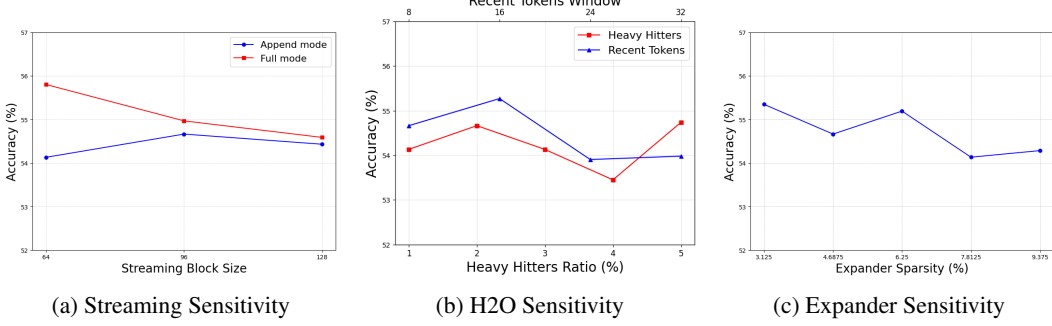

(a) Streaming Sensitivity  (b) H2O Sensitivity  (c) Expander Sensitivity

Figure 3: Hyperparameter studies of HEX with Llama-3 8B on GSM8k under 4-bit compression.

