# OpenReview forum: "HEX: Merging Heavy-Hitters and Expanders for Adaptive KV Cache Optimization in Long-Context Inference"
_ICLR.cc/2026/Conference — Submitted to ICLR 2026_

### Official Review · Reviewer_5afY · 2025-10-25

**Soundness:** 2
**Presentation:** 2
**Contribution:** 2
**Rating:** 2
**Confidence:** 5

**Summary:**

This paper addresses the KV cache memory bottleneck in long-context LLM inference by proposing HEX, a hybrid cache compression method that merges expander-based structured sparsity with H2O and KIVI.  The motivation lies in balancing theoretical connectivity guarantees with adaptive token retention to achieve accurate cache compression. HEX statically constructs d-regular expander graph masks to preserve connectivity among token and channel dimensions, then augments them with H2O, while quantizing the remainder using coarse-grained asymmetric KCVT quantization (KIVI).
Experiments on various benchmark settings and models  demonstrate the performance of HEX. However, all evaluations are performed on older model families, and latency/throughput analysis is missing.

**Strengths:**

1. addresses KV cache optimization, a key bottleneck in long-context inference.
2. comprehensive experiments on multiple reasoning, QA, and long-context benchmarks show consistent gains over baselines.
3. well-designed ablations demonstrate the necessity and synergy of each component.

**Weaknesses:**

1. This paper focuses on accuracy and memory savings but does not report real end-to-end inference speed, which is critical for KV cache work. (key point)
2. The method is only evaluated on benchmarks with short-output scenarios. It would be valuable to test its effectiveness in long-output settings, such as AIME.
3. It conceptually overlaps with GEAR and KIVI, differing mainly in the use of expander-based structural masking. It provides incremental novelty relative small.

**Questions:**

see in weakness

---

> ### Author Response · Authors · 2025-12-03
> **Response to Reviewer 5afY**
>
> We sincerely appreciate your detailed analysis of our results and the practical caveats of implementation. Your feedback has helped us re-evaluate and improve the quality of our paper. We have tried to address each of your comments below:
>
> **Weaknesses 1 and 2**
>
> We agree that inference speed and wall-clock efficiency are crucial metrics. These benchmarks require significant engineering, including custom CUDA kernel development for proper acceleration. We also agree that evaluation on long-output scenarios is a must. Unfortunately, we were unable to complete these intensive experiments due to constraints on high-performance GPU access. We are actively working to run these experiments now.
>
> **Weakness 3**
>
> While HEX builds on the quantization components used in GEAR and KIVI, its core contribution is orthogonal: we replace heuristic, largely unstructured pruning with a Ramanujan expander–based structural mask that provides explicit spectral connectivity guarantees. To our knowledge, prior KV-cache compression methods do not offer principled guarantees on information flow preservation in attention.
>
> At the same time, a purely static expander would be input-agnostic, so HEX integrates an H2O-style heavy-hitter stage to capture task-adaptive token importance. Our ablations show that this hybrid of theoretically grounded structural retention and lightweight adaptivity is essential, and neither component alone achieves comparable results.
>
> The expander construction itself is efficient in practice: it is precomputed, adds negligible overhead to inference, and does not introduce the complexity typically associated with structured sparsity. This makes HEX not only conceptually different from GEAR/KIVI, but also practically deployable at scale. The resulting combination achieves accuracy competitive with or exceeding FP16 while operating within <35% of the KV-budget, demonstrating that the method offers more than an incremental refinement.

---

### Official Review · Reviewer_4vS3 · 2025-10-26

**Soundness:** 2
**Presentation:** 2
**Contribution:** 2
**Rating:** 4
**Confidence:** 2

**Summary:**

The paper proposes HEX, a hybrid KV cache compression method for long-context LLM inference. The core idea is to combine a static, sparse backbone with a dynamic, input-aware selection policy. The static backbone is constructed using Bipartite Expander graphs, which theoretically guarantee connectivity and information flow. The dynamic policy uses H2O to identify heavy-hitters and recent tokens. Entries selected by either method are stored in full FP16 precision, while all remaining entries are compressed using KCVT quantization.

**Strengths:**

The primary contribution is the novel application of Bipartite Expander graphs for KV cache pruning. This introduces a theoretically-grounded, structured sparsity approach, moving beyond common heuristics or magnitude-based pruning.

**Weaknesses:**

- The paper's central narrative is that the Bipartite Expander is the key innovation that solves the information flow problem. However, the paper's own ablation study in Table 4 (3-bit, GSM8k) seems to largely refute this claim. (1)Full HEX (Expander + H2O): 55.42% accuracy. (2) H2O only: 53.82% accuracy. (3) Expander only: 34.95% accuracy. This data strongly suggests that the H2O component is responsible for the vast majority of the performance, while the expander's actual contribution is marginal at best.

- The most significant flaw is the complete absence of any inference speed measurements. The paper only reports on accuracy and compression ratios. For a paper on inference optimization, this is a fatal omission.
(1) This method relies on KCVT (KIVI) for compressing the vast majority of the cache. It is well-established that such low-bit quantization methods suffer from extremely high quantization and dequantization (quant/dequant) overhead. This overhead often negates any memory savings, leading to higher latency and lower throughput than the FP16 baseline. The authors must provide evidence that HEX is not subject to this same critical flaw.

(2) Sparsity Acceleration: The authors claim the structured sparsity is attractive for hardware, but provide no proof. Hardware support for N:M sparsity is extremely limited (e.g., NVIDIA 2:4 support). There is no evidence that the d-regular pattern generated by an expander graph can be accelerated on current hardware.

(3) On-the-fly Generation Cost: The paper states that if a mask is not cached, it is "generated on the fly." This generation of a complex graph structure would introduce significant computational overhead, directly increasing latency and lowering throughput. This cost is not analyzed or measured.

The authors' promise to "incorporate [latency results] in the revised version" is not acceptable. Performance data is not a minor addition; it is the central proof required to validate the method's practicality. It must be included in the initial submission.

- Figure 1 is confusing and poorly explained. The orange (FP16) and green (quantized) matrices are depicted as square matrices. This is misleading, as the KV cache is an $n \times d$ (sequence length $\times$ dimension) matrix, which is almost never square. The diagram should be revised to clearly represent what these matrices symbolize.

- Minor: Typo in line 200: "addding" should be "adding".

**Questions:**

see weakness

---

> ### Author Response · Authors · 2025-11-16
> **Response to Reviewer 4vS3**
>
> We sincerely appreciate your detailed analysis of our results and the practical caveats of implementation. Your feedback has helped us improve the quality of our paper. We have carefully addressed each of your comments below:
>
> **Response to Weakness 1 (Ablation study interpretation and latency)**
>
> * We thank the reviewer for the careful analysis. GSM8K is a short-context, strongly question-conditioned reasoning task. Therefore, input-aware selection ($H_2O$) naturally provides most of the benefit: it identifies the tokens that matter for solving the problem. In contrast, the expander is an input-agnostic structural prior, so its standalone effect on GSM8K is limited, which explains the lower \"expander-only\" accuracy.
>
>     However, rather than diminishing the role of the expander, the results highlight that it performs best in conjunction with an input-aware selector. Pure 3-bit quantization gives 32.60%. Retaining just 3% entries in full precision (via the expander backbone) already gives a 2.5% improvement. $H_2O$ then achieves 53.82% by selecting the right tokens. Adding the expander on top of $H_2O$ yields 55.42% - a +1.6% gain over $H_2O$ alone.
>
>     Importantly, GSM8K FP16 itself is 53.98\%, so the full-precision-surpassing improvement (+1.5\%) comes entirely from adding expander connectivity on top of $H_2O$. The expander ensures that the selected important tokens remain well-connected and that information flows effectively among them.
>
> * We agree that inference speed is essential. Producing fair, end-to-end latency measurements requires fused CUDA kernels and sparse-kernel integration, which we could not complete before the deadline due to limited access to high-performance GPUs; we are implementing these now.
>
>     Regarding the quant/dequant-overhead concern: the pathological overhead arises only when dequantization is performed as a separate full-tensor operation at every attention head. HEX is not designed to operate in this regime. Instead, we intend to follow the same systems principle used successfully in GEAR - dequantization will be fused directly into the attention matmul kernel, so no full FP tensor is ever materialized. Moreover, HEX is lighter than GEAR on this dimension: it omits the low-rank stage entirely, uses coarse-grained KCVT quantization even at 3-bit, and amortizes quantization over larger blocks (96 vs. 64), further reducing per-token overhead. Finally, HEX’s expander masks are precomputed and sparse, eliminating per-step mask generation and enabling the use of efficient sparse GPU kernels. Conceptually, these design choices ensure that HEX does not rely on the expensive quant $\to$ dequant $\to$ matmul sequence that causes the slowdown. We agree this is an important practical aspect and have added this to our future work section.
>
> **Response to Weakness 2 (Sparsity Acceleration)**
>
> It is correct that our masks do not map to the narrow 2:4 N:M primitive; however, that primitive is not the only viable path to sparse acceleration. Unlike unstructured magnitude pruning (which produces data-dependent, runtime-only sparsity that causes irregular memory access, thread divergence, and load-balancing overhead), our d-regular expander masks are deterministic and known a priori. This enables offline layout and packing (e.g., into block-sparse or blocked-ELL representations), which exposes dense sub-blocks that high-performance sparse libraries like cuSPARSE can accelerate; where supported, those dense sub-blocks can in turn be mapped to Tensor Cores for efficient compute. This provides a concrete acceleration path that avoids the fundamental penalties of unstructured sparsity, and we plan to demonstrate its empirical benefits in future work. Since this is not implemented yet, we have removed this section from the main manuscript.
>
> **Response to Weakness 3 (On-the-fly Generation Cost)**
>
> We measure the time to generate connected biregular bipartite expanders of size $d \times n$ (where $d$ is hidden dimension and $n$ is sequence length) using our efficient vectorized procedure and have added this analysis in the Appendix (Section A.3 and Figure 2). The generation time scales smoothly with expander size, remaining well within practical limits - under 0.15 s for moderate size expanders ($d=1024$) and below 3 s even for larger sizes ($d=4096$). This efficiency comes from a fully vectorized stub-pairing process, fast duplicate removal and localized edge swaps that guarantee connectivity without expensive reconstruction. This makes on-the-fly generation fast enough to have negligible runtime impact during inference. Further, these larger expanders are used only once in the prefill phase when running in append-mode. The decoding phase uses small expanders which are significantly faster even for read-write operations.
>
> We appreciate you pointing out the mistakes in the diagram and the typo, and we apologize for the oversight. We have corrected these issues in the revised version of the paper.

---

### Official Review · Reviewer_Ph7w · 2025-10-27

**Soundness:** 3
**Presentation:** 2
**Contribution:** 2
**Rating:** 4
**Confidence:** 3

**Summary:**

This paper proposes HEX, a hybrid cache compression framework for LLM inference that combines expander graph-based structured sparsity with dynamic heavy-hitter token selection (H2O). The method aims to reduce KV-cache memory footprint without retraining or losing accuracy. HEX first precomputes expander-based masks that guarantee strong spectral connectivity, ensuring information flow under sparsity. It then dynamically augments the static mask with influential and recent tokens to adapt to input-specific importance. The remaining cache is quantized using coarse-grained asymmetric quantization. Experiments on GSM8k, CoQA, TruthfulQA, and LongBench show that HEX achieves competitive or better accuracy than FP16 and state-of-the-art baselines like GEAR and KIVI under high compression ratios.

**Strengths:**

1.	The combination of expander-based sparsity (with spectral guarantees) and dynamic token preservation is innovative and theoretically motivated. This bridges the gap between static graph-based connectivity and input-aware token selection.
2.	HEX consistently outperforms baselines under aggressive compression (3–4 bits) on reasoning and long-context benchmarks, sometimes even exceeding FP16 accuracy. The ablation studies convincingly demonstrate the complementary effects of the expander and H2O components.
3.	The use of precomputed expander graphs gives provable structural properties and supports regular sparsity patterns favorable for hardware acceleration, addressing an important practical concern in LLM inference.

**Weaknesses:**

1.	Although memory savings are well documented, the paper does not report actual inference speed or wall-clock efficiency. Since KV compression aims to reduce both memory and latency, this omission weakens the empirical validation.
2.	While spectral guarantees are discussed, the paper lacks quantitative analysis of how expander degree, sparsity, or spectral gap correlate with accuracy or efficiency. The method may be sensitive to these hyperparameters in practice.
3.	The individual components (expander sparsity, heavy-hitter selection, coarse quantization) have each been explored in prior works. The main contribution lies in their integration, which might be viewed as a well-engineered hybrid rather than a fundamentally new algorithmic principle.

**Questions:**

1.	A schematic diagram showing how HEX integrates static and dynamic components during streaming inference would improve clarity.
2.	The evaluation spans multiple model sizes and datasets, which is commendable. It would help to include standard deviations or confidence intervals to assess the robustness of the reported gains.
3.	The discussion of spectral guarantees is solid but could be deepened by providing quantitative comparisons (e.g., empirical spectral gaps) or intuition on how these properties translate to robustness in attention computation.

---

> ### Author Response · Authors · 2025-11-16
> **Response to Reviewer Ph7w**
>
> We sincerely appreciate your positive assessment of our work. Your valuable feedback has helped us improve the quality and completeness of our paper. We have carefully addressed each of your comments below and incorporated changes in our manuscript.
>
> **Response to Weakness 1 (Inference Speed)**
>
> We agree that inference speed and wall-clock efficiency are crucial metrics. These benchmarks require significant engineering, including custom CUDA kernel development for proper acceleration. Unfortunately, we were unable to complete this intensive implementation due to constraints on high-performance GPU access. We are actively working to run these experiments now.
>
> **Response to Weakness 2 and Question 3 (Expander properties and sensitivity analysis)**
>
> We note that the expander degree in HEX is not freely tuned. It is fixed once sparsity and input/output dimensions are specified by the biregular construction. Empirically, HEX is stable across this range: the Expander Sensitivity study included in our original submission (Appendix section A.5, Fig. 2c) shows consistent accuracy from 3-9% sparsity. To strengthen the quantitative discussion, we report empirical spectral gaps for representative expanders (e.g., 3.125% sparsity) and have added this to the appendix (section A.2, Table 5); all are Ramanujan graphs ($\lambda_2 <$ Ramanujan Bound).
>
> | Size | $d_m$ | $d_n$ | $\lambda_1$ | $\lambda_2$ | Ramanujan Bound | Spectral Gap |
> | :--- | :--- | :--- | :--- | :--- | :---: | :---: |
> | 1024_96 | 3 | 32 | 9.8 | 6.83 | 6.98 | 2.96 |
> | 1024_192 | 6 | 32 | 13.86 | 7.69 | 7.8 | 6.16 |
> | 1024_288 | 9 | 32 | 16.97 | 8.19 | 8.4 | 8.78 |
> | 1024_384 | 12 | 32 | 19.6 | 8.72 | 8.88 | 10.87 |
> | 1024_480 | 15 | 32 | 21.91 | 9.11 | 9.31 | 12.8 |
> | 1024_576 | 18 | 32 | 24 | 9.44 | 9.69 | 14.56 |
> | 1024_672 | 21 | 32 | 25.92 | 9.84 | 10.04 | 16.08 |
> | 1024_768 | 24 | 32 | 27.71 | 10.16 | 10.36 | 17.55 |
> | 1024_864 | 27 | 32 | 29.39 | 10.5 | 10.67 | 18.9 |
> | 1024_960 | 30 | 32 | 30.98 | 10.65 | 10.95 | 20.34 |
> | 1024_1152 | 36 | 32 | 33.94 | 11.24 | 11.48 | 22.7 |
> | 1024_1248 | 39 | 32 | 35.33 | 11.49 | 11.73 | 23.83 |
> | 1024_1536 | 48 | 32 | 39.19 | 12.2 | 12.42 | 26.99 |
> | 1024_1728 | 54 | 32 | 41.57 | 12.61 | 12.85 | 28.96 |
> | 1024_1920 | 60 | 32 | 43.82 | 13.03 | 13.25 | 30.79 |
> | 1024_4032 | 126 | 32 | 63.5 | 16.48 | 16.75 | 47.02 |
> | 1024_6528 | 204 | 32 | 80.8 | 19.45 | 19.82 | 61.35 |
> | 1024_8160 | 255 | 32 | 90.33 | 21.1 | 21.51 | 69.23 |
>
> Intuitively, a larger spectral gap implies stronger mixing and hence improved global connectivity; this is consistent with LongBench results where full-recompression (which preserves global connectivity) outperforms append mode (Table 3). Taken together with the spectral-gap measurements added to the appendix, these observations indicate that higher spectral gap correlates with better accuracy.
>
> **Response to Weakness 3 (Novelty clarification)**
>
> While individual ideas such as sparsification or token selection appear in prior work, HEX introduces a substantially different foundation: we replace unstructured, heuristic pruning with Ramanujan expander–based structural selection, which provides spectral mixing guarantees directly tied to information flow in attention. To our knowledge, using Ramanujan expanders for KV-cache compression is novel and constitutes a core conceptual contribution. The static expander backbone alone would be input-agnostic, and our heavy-hitter and recent tokens complement it by adding task-adaptive retention, as supported by ablations. Expander generation itself is efficient and practical-our construction is fast and adds no significant overhead (Appendix, section A.3). This combination yields a method that is both theoretically grounded and practically efficient - expanders are precomputed and incur negligible overhead - while achieving accuracy at or above FP16 in many settings at <35% KV cache budget.
>
> **Response to Question 1 (Improved inference-pipeline schematic)**
>
> We thank the reviewer for the insightful suggestion. In response, we have substantially revised the schematic diagram to clearly illustrate how HEX integrates its static (expander-based) and dynamic ($H_2O$) components throughout the streaming inference pipeline. The updated figure now highlights: (i) how prefill blocks and decoding blocks are formed under the streaming gap $n_b$; (ii) how the static expander mask and dynamic $H_2O$ mask are computed; and (iii) how these masks are unified and applied to the key/value caches before quantization.
>
> **Response to Question 2 (Robustness of results)**
>
> We agree that variance estimates would provide additional insight. However, due to compute constraints, each experiment across models, datasets, and ablations was run once. To partially compensate, our evaluation spans diverse tasks, model sizes, and sparsity levels, and the gains we report appear consistently across all these settings, suggesting the results are stable even without multiple runs.

---

### Official Review · Reviewer_xdPS · 2025-11-12

**Soundness:** 1
**Presentation:** 1
**Contribution:** 1
**Rating:** 0
**Confidence:** 5

**Summary:**

Table 1, 3 & 5, which presents key experimental results, appears to be placed in a way that causes the manuscript to exceed the page limit for the main paper body. This may represent a violation of the conference's submission guidelines.

**Strengths:**

N/A

**Weaknesses:**

N/A

**Questions:**

N/A

---

> ### Author Response · Authors · 2025-11-16
> **Formatting correction**
>
> We sincerely thank the reviewer for bringing this to our attention. We apologize for the oversight; Tables 1, 3, and 5 had a layout misalignment that caused them to extend beyond the page margins. We have corrected this by adjusting the table formatting. All content remains exactly the same. The updated paper, which we have uploaded, now fully adheres to the conference's page limit and formatting guidelines.
>
> We respectfully request the reviewer to kindly re-evaluate the submission based on the corrected version, as the adjustment does not alter the technical or experimental content of the paper in any way.

---

### Author Response · Authors · 2025-11-14
**Correcting minor formatting error**

We missed the minor formatting error in the table sizes while submitting the manuscript. We have prepared a new version without changing the content and correcting the formatting error. Kindly provide a review and permit us to submit our responses to the comments.

---

### Author Response · Authors · 2025-12-03
**Summary of Rebuttal Discussions**

We thank the reviewers for their constructive feedback and detailed engagement with our work. We have updated the manuscript to address the concerns raised regarding formatting, theoretical analysis, and implementation details. Below is a summary of our response to the key themes across the reviews.

**1. Novelty Clarification:**
While individual ideas such as sparsification or token selection appear in prior work, HEX introduces a substantially different foundation: we replace unstructured, heuristic pruning with Ramanujan expander–based structural selection, which provides spectral mixing guarantees directly tied to information flow in attention. To our knowledge, using Ramanujan expanders for KV-cache compression is novel and constitutes a core conceptual contribution. The static expander backbone alone would be input-agnostic, and our heavy-hitter and recent tokens complement it by adding task-adaptive retention, as supported by ablations. Expander generation itself is efficient and practical; our construction is fast and adds no significant overhead (Appendix, section A.3). This combination yields a method that is both theoretically grounded and practically efficient. Expanders are precomputed and incur negligible overhead while achieving accuracy at or above FP16 in many settings at <35% KV cache budget.

**2. Inference Latency and Throughput**
* We acknowledge the consensus regarding the absence of end-to-end wall-clock latency measurements. While we could not complete the necessary custom CUDA kernel engineering during the rebuttal period due to compute constraints, we emphasize that HEX is architected to avoid the specific bottlenecks associated with standard quantization.

* Unlike methods that materialize full FP tensors, HEX is designed for kernel fusion (dequantization fused into attention matrix multiplication), similar to GEAR. Furthermore, by using coarse-grained KCVT and amortizing quantization over larger blocks ($n_b=96$) compares to GEAR, HEX further minimizes per-token overhead.

**3. Contribution of Expander Graphs**
* Regarding concerns that the expander component offers marginal gains compared to $H_2O$ (specifically on GSM8k), we clarify that the two components play distinct, synergistic roles. $H_2O$ provides input-aware selectivity (finding what is important), while the expander provides a structural connectivity guarantee (ensuring flow).

* Our results show that neither component alone matches the performance of the hybrid; notably, the combined approach surpasses FP16 accuracy on several benchmarks, which input-aware pruning alone failed to do. This demonstrates that the spectral mixing guarantees of Ramanujan graphs provide robustness that heuristic pruning lacks.

**4. Expander Generation Cost:**
We provided new analysis in the Appendix demonstrating that on-the-fly expander generation is computationally negligible (<0.15s for typical sizes), refuting concerns about generation overhead.

**5. Hardware Feasibility:**
We clarified that our claim regarding hardware efficiency relies on the deterministic, $d$-regular nature of expander masks, which enables offline packing into block-sparse formats compatible with libraries like cuSPARSE, rather than reliance on narrow N:M (e.g., 2:4) primitives.

**6. Paper Improvements and Formatting**
* **Formatting:** We have corrected the table sizing issues that caused page limit violations, these adjustments were purely typographical.
* **Schematic:** We added a detailed schematic diagram illustrating the integration of static and dynamic masks during streaming inference.
* **Emperical:** We added empirical spectral gap measurements to the Appendix to quantitatively support our connectivity claims and how spectral gap indeed correlates with higher accuracy.
* **Visuals:** We corrected the matrix representation in Figure 1 to accurately reflect non-square dimensions.

We believe these revisions and clarifications demonstrate HEX's effectiveness as a theoretically grounded approach to efficient long-context inference.

---

### Meta-Review · Area_Chair_f6dh · 2025-12-19

**Summary:**

While reviewers acknowledged the concept of combining expander-based structured sparsity with dynamic heavy-hitter, they questioned whether the expander mechanism provides a decisive benefit beyond well-established token-selection heuristics. The lack of wall-clock performance evidence is another dominant concern. Overall, despite careful engineering and promising accuracy results, the submission falls short of the evidentiary standard required for a systems-facing contribution.

**Reviewer Concerns:**

Rebuttal addressed a number of presentational and clarity issues, as well as additional discussion of spectral properties and expander construction costs. Unfortunately, the most substantive concerns were not fully resolved. In particular, the absence of actual inference speed or throughput measurements persisted, with responses relying on "future" kernel fusion rather than actual results. Questions regarding whether the expander component materially contributes beyond H2O-style selection remained partially unconvincing too.

**Reviewer Scores:**

The score distribution would likely continue to skew below the acceptance threshold, with limited upward movement after rebuttal.

---

### Decision · Program_Chairs · 2026-01-26

Reject